

# Dynamic frequent subgraph mining algorithms over evolving graphs: a survey

Belgin Ergenç Bostanoğlu and  Nourhan Abuzayed

Computer Engineering, Izmir Institute of Technology, Izmir, Turkey

## ABSTRACT

Frequent subgraph mining (FSM) is an essential and challenging graph mining task used in several applications of the modern data science. Some of the FSM algorithms have the objective of finding all frequent subgraphs whereas some of the algorithms focus on discovering frequent subgraphs approximately. On the other hand, modern applications employ evolving graphs where the increments are small graphs or stream of nodes and edges. In such cases, FSM task becomes more challenging due to growing data size and complexity of the base algorithms. Recently we see frequent subgraph mining algorithms designed for dynamic graph data. However, there is no comparative review of the dynamic subgraph mining algorithms focusing on the discovery of frequent subgraphs over evolving graph data. This article focuses on the characteristics of dynamic frequent subgraph mining algorithms over evolving graphs. We first introduce and compare dynamic frequent subgraph mining algorithms; trying to highlight their attributes as increment type, graph type, graph representation, internal data structure, algorithmic approach, programming approach, base algorithm and output type. Secondly, we introduce and compare the approximate frequent subgraph mining algorithms for dynamic graphs with additional attributes as their sampling strategy, data in the sample, statistical guarantees on the sample and their main objective. Finally, we highlight research opportunities in this specific domain from our perspective. Overall, we aim to introduce the research area of frequent subgraph mining over evolving graphs with the hope that this can serve as a reference and inspiration for the researchers of the field.

Corresponding author
Belgin Ergenç Bostanoğlu,
belginergenc@iyte.edu.tr

# INTRODUCTION

Graphs are used in many real-world data sets since they allow expressing various relationships between nodes; *e.g.*, relationship (edges) between users (nodes) in social networks, bonds (edges) between atoms (nodes) in chemical structures (*Chakrabarti & Faloutsos, 2006*; *Fournier et al., 2020*). Graph mining became more popular since the data represented by graphs increased and the need of knowledge discovery from this type of data became important. Graph mining has several sub-categories such as graph classification, graph clustering and frequent subgraph mining (*Jiang, Coenen & Zito, 2013*) and these can be used in different domains *e.g.*, social media analysis and biological research.

Frequent subgraph mining (FSM) can be expressed as discovering all the subgraphs that are found in a graph more than a given support threshold. Recurrent structures can show themes or ideas related in the given graph database and can be used further for performing other graph mining applications such as graph classification, graph partitioning, graph clustering, graph correlations *etc* (*Cuzzocrea et al., 2015*). We see various FSM algorithms designed for undirected graphs (*Yan & Han, 2002*; *Huan, Wang & Prins, 2003*) directed graphs (*Sangle & Bhavsar, 2016*) or large graphs (*Bhatia & Rani, 2018*). Some of these algorithms are Apriori based (*Inokuchi, Washio & Motoda, 2000*) wheras the others are pattern growth based (*Kuramochi & Karypis, 2005*). The objective of these algorithms also change as to find all frequent subgraphs (*Huan, Wang & Prins, 2003*), approximate frequent subgraphs (*Preti, De Francisci Morales & Riondato, 2023*), closed frequent subgraphs (*Yan & Han, 2003*) or maximal frequent subgraphs (*Huan et al., 2004*). There are detailed qualitative surveys that compare FSM algorithms according to their various attributes (*Jiang, Coenen & Zito, 2013*; *Lakshmi & Meyyappan, 2012*; *Güvenoğlu & Bostanoğlu, 2018*).

Nowadays, dynamic graph-based applications which deal with emerging dynamic data; *i.e.,* social networks where friendships (*i.e.,* edges of graph) are linked and dissolved over time, protein-to-protein interaction networks where knowledge is frequently updated. Because of these applications, the need for incremental frequent subgraph mining approaches has become a necessity. The increments can be either (a) a series of small graphs, or (b) a stream of node and edge updates to the graph (*Ray, Holder & Choudhury, 2014*). These increments can be (i) edges or/and nodes are to be added to the graph over time, (ii) attributes of existing edges or/and nodes to be modified over time, (iii) edges or/and nodes that are present and to be removed from the graph.

Although frequent subgraph mining has been widely studied few works exist for dynamic frequent subgraph mining (*Kuramochi & Karypis, 2004*; *Abdelhamid et al., 2017*). Static algorithms assume that graphs do not change over time and try to find all frequent subgraphs in the data. On the other hand, dynamic frequent subgraph mining algorithms deal with change in the data. Most of these algorithms concentrate on exact output where the idea is to find all frequent subgraphs overall input data at a specific timestamp. Exact algorithms search for all the frequent patterns; this requires high execution time and memory consumption. Therefore, for faster results users are willing to trade-off accuracy in cases where approximate results can serve the purpose.

There are few recent works that are designed for approximate outputs on dynamic graphs like Triest (*De Stefani et al., 2016*), SR and OSR (*Aslay et al., 2018*) and TipTap collection (*Nasir et al., 2021*). They provide simple approximate approaches with trade-off between time and accuracy. When these algorithms are analyzed, in addition to the characteristics of dynamic subgraph mining algorithms we see aspects related with their sampling schemes and objectives. They use either fixed size sample or flexible sample size that changes according to desired output accuracy level. Some of these algorithms have the objective of minimizing execution time while the others focus on maximizing the accuracy of the results.

## Scope and motivation of the survey

When we scan the literature related with algorithms on dynamic subgraph graph mining over dynamic graphs we come across few surveys (*Fournier et al., 2020*; *Chaudhary & Thakur, 2018*). They concentrate on discovering frequent subgraphs, evolution rules, motifs, subgraph sequences, recurrent and triggering patterns, and trend sequences on graphs where dynamic aspect can be incremental based, snapshot-based or interval based. Our motivation on the other hand is to be able present a comprehensive and focused literature review on "dynamic frequent subgraph mining algorithms" only. Another distinction of our survey from the previous surveys we consider only "evolving graphs" with increments either as a series of small graphs or stream of nodes and edges.

We try to present and compare frequent subgraph mining algorithms over evolving graphs. We analyze them under two main groups where the first group is exact algorithms whose objective is to find all frequent or closed subgraphs of the evolving graph and the second groups is approximate algorithms whose objective is to discover frequent subgraphs by employing a sampling strategy.

In summary, this article makes following contributions:

- It gives detailed background information related with graph basics, increment types of dynamic graphs, frequent subgraph mining, dynamic frequent subgraph mining and sampling strategies in the context of approximate dynamic frequent subgraph mining.

- It gives a qualitative literature survey on dynamic frequent subgraph mining algorithms designed for evolving graphs. Review contains a comparison of these algorithms with respect to their various characteristics. In addition, each of them is introduced briefly.

- It gives another qualitative literature survey on approximate dynamic subgraph mining algorithms again with respect to their various characteristics. This time the characteristics include the ones related with their sampling strategy as well. Similar to the previous part, the review contains a comparison of the algorithms in addition to the brief introduction of each.

- It tries to contribute both as an introduction and as a chronological guide to recent advances and opportunities in this specific research area of frequent subgraph mining over evolving graphs.

## Outline of the survey

The rest of this survey flows as follows: In "Survey Methodology" section, we address our research questions, data sources and research strategy, inclusion/exclusion criteria. In the "Background" section, we introduce the preliminary information required in explaining and comparing the algorithms of the survey. In the "Frequent Subgraph Mining Algorithms for Evolving Graphs" section, we evaluate dynamic frequent subgraph mining algorithms within the scope of the survey with a detailed comparison and discussion. In "Approximate Frequent Subgraph Mining Algorithms for Evolving Graphs" section we outline approximate frequent subgraph mining algorithms again with a comparative perspective as well. In "Research Opportunities" section, we try to highlight potential future research directions in the field. Finally, in "Conclusion" section, we summarize all our findings in the survey.

Portions of this text were previously published as part of a thesis (https://openaccess.iyte.edu.tr/bitstream/11147/12616/1/10147615.pdf)'.

## SURVEY METHODOLOGY

As indicated in the last part of the "Introduction" section our motivation for this survey is to produce a focused and detailed literature review on exact and approximate frequent subgraph mining algorithms devised for evolving graphs. In this section, we give our research methodology starting by introduction of our research questions in mind. We continue by explaining the data sources and research strategy used in the survey. We will end the section by talking about our inclusion/exclusion criteria related within the field of subgraph mining over dynamic graphs.

Our literature review aims to address following research questions:

- RQ1: What are the types of increments handled by dynamic frequent subgraph mining algorithms over evolving graphs?
- RQ2: What are the sampling strategies used in dynamic frequent subgraph mining algorithms over evolving graphs?
- RQ3: What are the differences between exact dynamic frequent subgraph mining algorithms over evolving graphs?
- RQ4: What are the differences between approximate dynamic frequent subgraph mining algorithms over evolving graphs?
- RQ5: What are the potential research opportunities in the context of dynamic frequent subgraph mining over evolving graphs?

We discuss the issues related to the first and second questions in the "Background" section. The third and fourth, questions are detailed in "Frequent Subgraph Mining Algorithms for Evolving Graphs" and "Approximate Frequent Subgraph Mining Algorithms for Evolving Graphs" sections consecutively. The last research question is answered by using our findings after our attentive effort of completing this focused survey.

### Data sources and research strategy

We utilized Google Scholar, ACM Digital Library, IEEE Explore, Wos, Arxiv, Siam Journal and Springer Link in finding related literature. The search strings that we used in the search are "frequent subgraph mining" AND "evolving graph" AND "approximate subgraph discovery" AND "frequent patterns" AND "dynamic networks" in all the search engines. Table 1 presents all the data sources, links and the search strings used in data collection. Same in the second column means that the search strings are the same as indicated in the first row of the table.

### Criteria for inclusion/exclusion

After searching all related literature from different search engines, we include only (i) research articles written in English, (ii) research articles published between the years 2006–2024, and (iii) research articles received from peer-reviewed resources. We exclude (i) parallel algorithms of the field and (ii) algorithms focusing on complex graphs such as attributed graphs.

**Table 1    The description of data sources, search strings and links.**

| Search Engine | Search string | Links |
|---|---|---|
| Google Scholar | "frequent subgraph mining" AND "evolving graph" AND ""approximate subgraph discovery" AND "frequent patterns" AND "dynamic networks" | https://scholar.google.com/ |
| ACM Digital Library | Same | https://dl.acm.org/ |
| IEEE Xplore | Same | https://ieeexplore.ieee.org/ |
| WoS | Same | https://webofknowledge.com/ |
| Arxiv | Same | https://arxiv.org/ |
| Siam Journal | Same | https://www.siam.org/ |
| Springer Link | Same | https://link.springer.com/ |

No parallel/only frequent subgraphs/no complex graph types/incremental not snapshot based not window based.

## BACKGROUND

In this section, we introduce the background information required in explaining and comparing the algorithms in the survey. We detail the graph basics, increment types of dynamic graphs, frequent subgraph mining process, dynamic frequent subgraph mining process and sampling strategies of approximate dynamic frequent subgraph mining algorithms.

### Graph basics

A **graph** is defined as a set of vertices (nodes) that are interconnected by a set of edges. Example of a graph is shown in Fig. 1. A graph G is an ordered pair (V, E) consisting of a set of vertices V = {v1, v2, v3, v4, v5} and V is connected to each other by and a set of edges E = {e1, e2, e3, e4}. A label function, $L$, maps a vertex or an edge to a label.

Assume **subgraph** G '(V', E') is a subgraph of the graph G (V, E), where edges and vertices are subsets of E and V respectively. Figure 2 shows examples of subgraphs such that S1, S2, and S3 are subgraphs of G of Fig. 1.

**Subgraph isomorphism**: Given two undirected graphs G and H. There is an isomorphism between G and H, if there is a bijection $f$ between their vertices ($f$: V(G) →V(H)), Hence, two vertices u, v are adjacent to each other in G if and only if $f$ (u), $f$ (v) are adjacent in H. These two graphs are topologically identical. In Fig. 3, the subgraph $g_1$ isisomorphic to the subgraph ($u_3$, $u_4$, $u_5$), where ($u_3$, $u_4$, $u_5$) is a subgraph of the input graph G. This is also called an **embedding** $g_1$ in graph in G.

**Static graph**: A graph G = (V, E) consists of a set of nodes V, and a set of edges E ⊆ V ×V, where V and E do not change over time.

**Dynamic graph (evolving graph)**: An evolving graph $G_D$ = ($V_D$, $E_D$) consists of a set of nodes $V_D$, and a set of edges $E_D$ ⊆ $V_D$ × $V_D$. $G_D$ is changed by node additions or deletions, edge additions or deletions over time. Figure 3 shows an example of a dynamic graph at

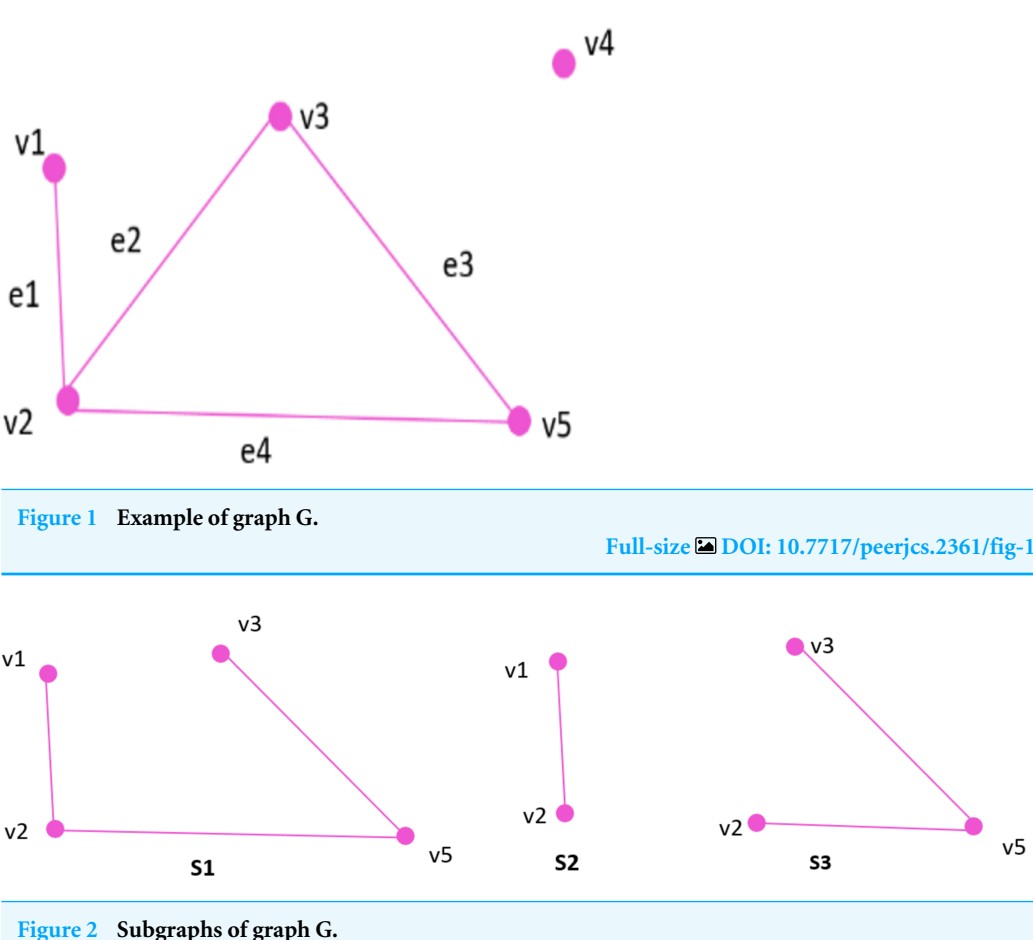

**Figure 1** Example of graph G.

**Figure 2** Subgraphs of graph G.

three points of time ($t_1$, $t_2$ and $t_3$). At time $t_2$, there is a deletion of the edge $u_{7-}u_8$. At time $t_3$, edge $u_{7-}u_8$ andedge $u_{3-}u_{10}$are added.

## Increments for dynamic graphs

In the context of dynamic graphs, graph increments can be represented as (a) a series of small graphs, or (b) a stream of node and edge updates (*Ray, Holder & Choudhury, 2014*).
**Increments as a stream of nodes and edges**: In this kind of dynamic graphs, the increments consist of nodes and edges that change over time, and it can be addition or deletion.

Dynamic graph (evolving graph): An evolving graph $G_D = (V_D, E_D)$ consists of a set of nodes $V_D$, and a set of edges $E_D \subseteq V_D \times V_D$. $G_D$ is changed by node additions or deletions, edge additions or deletions over time. Figure 3 shows an example of an edge updates, it shows the dynamic graph at different time points $t_1$, $t_2$ and $t_3$ when edges are coming as updates. At time $t_2$, there is a deletion of the edge $u_7$-$u_8$. At time $t_3$, edge $u_7$-$u_8$ and edge $u_3$-$u_{10}$ are added.
**Increments as a series of small graphs**: In this kind of dynamic graphs, the increments are done by using a series of graph objects, *i.e.,* each object in the stream is considered as a (static) graph snapshot.

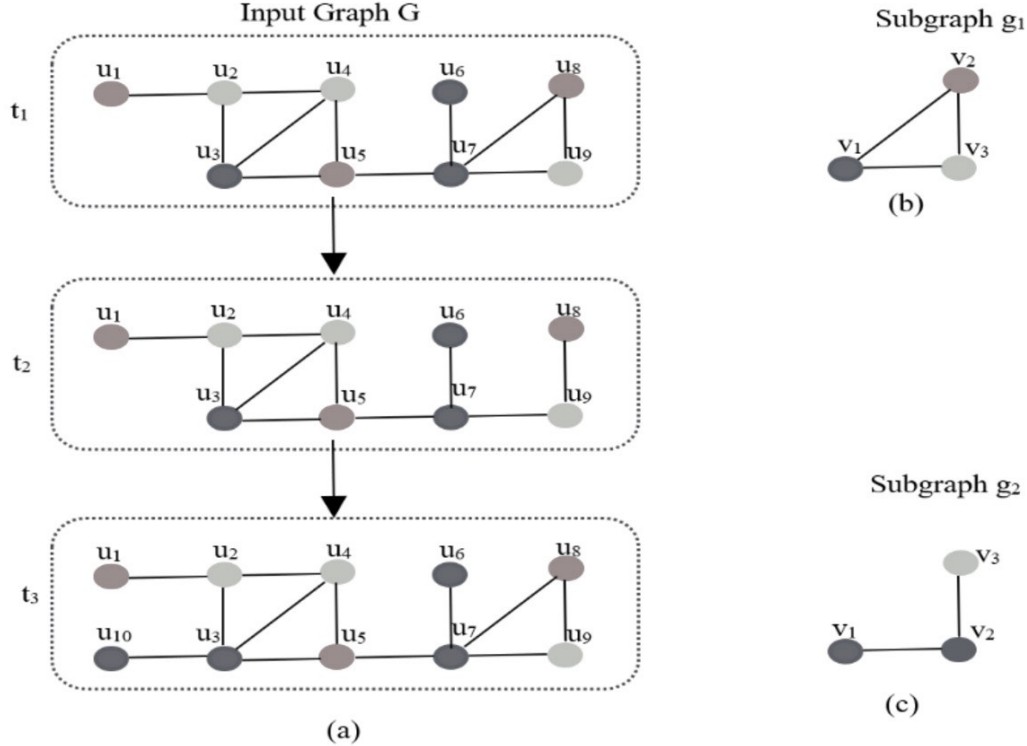

**Figure 3** Example of graph inputs as a series of small graphs.

Series of small graphs: Given a sequence $G_{ts}$ of n graphs $\{G_1, \ldots, G_n\}$ with $G_i = (V_i, E_i)$ for $1 \le i \le n$. We define $G_{ts}$ to be a time series of graphs if $V_1 = V_i$ for all $1 \le i \le n$. $G_i$ is the i-th state of $G_{ts}$ (*Borgwardt, Kriegel & Wackersreuther, 2006*).

Figure 4 shows an example of a dynamic graph when the input is a series of small graphs. It shows a graph input, where each incoming object is an entire graph.

## Frequent subgraph mining

The objective of a frequent subgraph mining (FSM) task is to find all frequent subgraphs in a given graph or a set of graphs that occur more than a given user-defined threshold (*Aggarwal & Wang, 2010*). Number of occurrences of a subgraph (g) in a graph dataset is called support of subgraph (g). If the support of subgraph is more than minimum support threshold that is defined by the user, then this subgraph is a frequent subgraph (*Dinari & Naderi, 2014*).

Figure 5 illustrates an example of finding frequent subgraphs. Input is a database of graph transactions, undirected simple graph (no loops, no multiples edges), each graph transaction has labels associated with its edges and vertices, transactions might not be connected and a minimum support threshold $\sigma$; example (60%). The output is frequent

| Transaction ID | Graph |
|:---:|:---:|
| 1 |  |
| 2 |  |
| 3 |  |
| 4 |  |
| 5 |  |
| 6 |  |

**Figure 4** (A) Dynamic graph G at different points of time (B) subgraph g1 (C) subgraph g2.

subgraphs that satisfy the minimum support threshold, and each frequent subgraph is connected.

Frequent subgraph mining task is computationally challenging since number of possible pattern experiences a combinatorial explosion with the maximum number of vertices and vertex labels. Finding isomorphic subgraphs is the other challenge of this process.
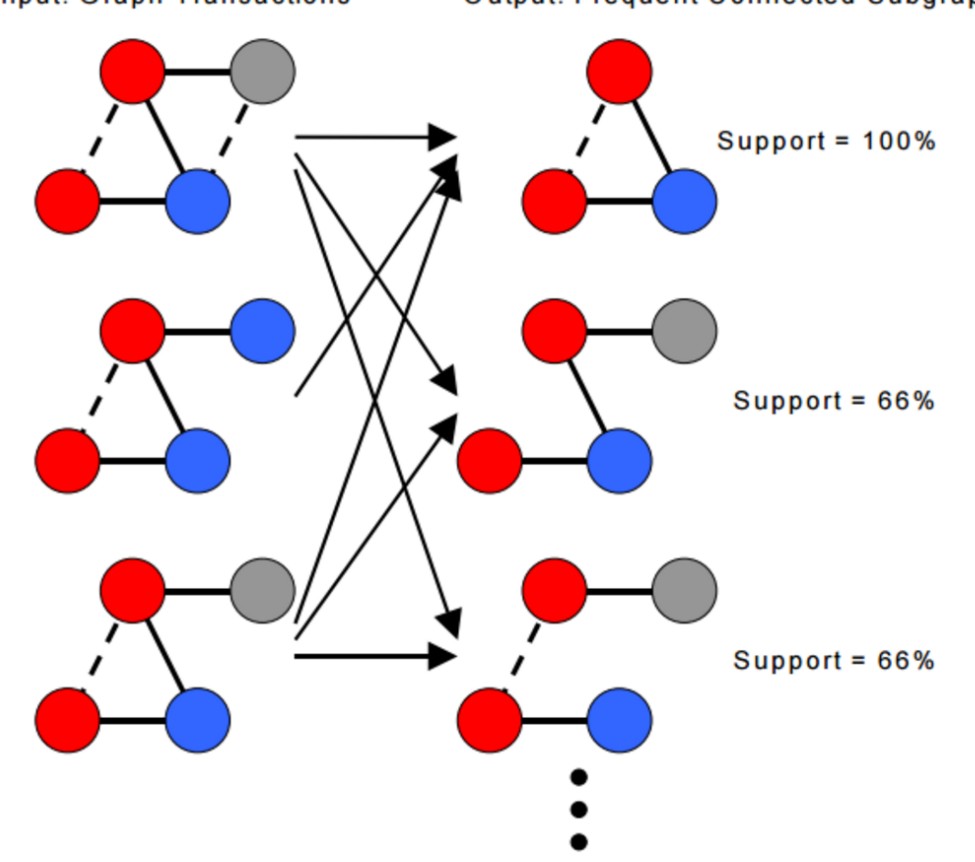

**Figure 5** Finding frequent subgraphs (input and output).

## Dynamic frequent subgraph mining

It is the process of finding all frequent subgraphs on evolving graph. In the example presented in Fig. 3 given an input dynamic graph G and support threshold 2, let us see the status of subgraphs $g_1$ and $g_2$. Addition of an edge to input graph increases the support of one or more subgraphs, removal of an edge of the input graph decreases the support of one or more subgraphs (*Abdelhamid et al., 2017*). At time $t_1$, the subgraph $g_1$ has two matches in G, however the subgraph $g_2$ has only one match. As a result, $g_1$ is frequent subgraph and $g_2$ is not frequent subgraph. At time $t_2$, there is a deletion of the edge $u_{7-}u_{8,}$ so the number of embeddings of $g_1$ decreases to one, however the number of embedding of $g_2$ does not change. Therefore, both subgraphs $g_1$ and $g_2$ are not frequent. At time $t_3$, there is an addition of edge $u_{3-}u_{10}$, this addition increases the number of matches of $g_2$ to two; so, $g_2$ becomes frequent.

## Graph sampling

Graph sampling is needed in network analysis, to keep the graph scale small while capturing the properties of the original graph, graph sampling provides an efficient, yet inexpensive solution for network analysis (*Wang et al., 2011*). Another reason for using graph sampling;

it is sometimes only the solution since obtaining all data for a graph is not permitted or is very time-consuming. Thus, we must obtain the properties of the graph by sampling.

Graph sampling can make the graph size smaller while keeping the characteristics of the original graph (*Sahu et al., 2023*; *Wang et al., 2011*). Graph sample of Graph G = (V, E) is defined as $G_s = (V_s, E_s)$ where $V_s \in V$, $E_s \in E$ and $E_s \subseteq \{(u, v) | u \in V_s,$ and $v \in V_s\}$. There are different sampling methods like Node Sampling, Edge Sampling, Traversal-based sampling (*Zhang et al., 2017*) and Sampling with Neighborhood (VSN) (*Hu & Lau, 2013*). Our intention is not give detail review of graph sampling strategies and methods but we would like to mention the ones used in the context of frequent subgraph discovery by static and dynamic frequent subgraph mining algorithms.

## Approximate dynamic subgraph mining

FSM task on large graphs is computationally challenging due to combinatorial explosion in possible number of vertex and vertex labels of subgraphs and subgraph isomorphism operations needed. This process is generally NP-complete and exact algorithms tend to scale poorly. Using a sample of the graph whenever the approximate results serve the purpose is a solution. Another reason for using sampling is for the cases when the entire graph cannot be retrieved. Main challenge of sampling is to provide theoretical guarantees on the quality of the sample. We see a good example sampling based randomized algorithm MaNIACS (*Preti, De Francisci Morales & Riondato, 2023*) for computing high quality approximations of the collection of the frequent subgraphs in a vertex-labeled graph. The upper bound of the estimation corresponding to maximum allowed estimation error relies on empirical Vapnik–Chervonenkis dimension (*Riondato & Upfal, 2018*). MaNIACs algorithm is designed for static graph and used for creating sample graph that is used in discovering approximate frequent subgraphs.

When we deal with approximate solutions in dynamic frequent subgraph mining we see four sampling algorithms Simple Random Sampling (*Yates, Moore & Starnes, 2002*), Neighborhood Sampling (*Ray, Holder & Choudhury, 2014*), Reservoir Sampling (*Vitter, 1985*) and Sampling using Vapnik–Chervonenkis dimension (*Nasir et al., 2021*).

**Simple random sampling**: *Yates, Moore & Starnes (2002)*: In this sampling scheme an item from the data stream is picked or not picked for the sample with a user defined probability. The inclusion probability can be given to the algorithm as an input parameter. For example if inclusion parameter is 0.5, then each item has an equal probability of being picked or not. It is a simple scheme but we can see two disadvantages: (i) size of the sample grows as the size of the input data stream grows and (ii) quality of the sample can only be guaranteed when the sample size exceeds specific size (*Anis & Nasir, 2018*).

**Neighborhood sampling**: *Ray, Holder & Choudhury (2014)*: We see this sampling approach in StreamFSM algorithm, which reports the frequent subgraphs that are estimated to be found in the entire graph as each batch, arrives. When a new edge arrives to the graph, a number of nodes that are one hop away from this edge point is selected. Then edges between these nodes are included in the sample. The idea is to restrict the changes to the set of frequent subgraphs to the locality of this new edge. Selecting the nodes, however, is dependent on the domain. Domains that have a high frequency of large star-shaped

patterns, where the edges in the star have similar labels, require some pruning. In other domains to select all the nodes in the immediate neighborhood of the edge is possible. These extracted neighborhoods now become graph transactions, which are input to a graph transaction miner like gSpan (*Yan & Han, 2003*). Selecting a higher number of neighbors increases the running time and the accuracy, while selecting a lower number of neighbors does the reverse.

**Reservoir sampling**: *Vitter (1985)*: It is a fixed-size randomized sampling scheme. It maintains a fixed-size uniform sample of the data stream as first introduced (*Gemulla, Lehner & Haas, 2006*). The size of the sample is provided as the input parameter. The algorithm initializes with a fixed-size sample. When the sample reaches to the size defined as input, the randomly chosen item from the data stream is added to the sample and an existing item from the sample is deleted. Therefore, the reservoir-of sample size is kept constant. This random pairing is a fully dynamic algorithm for reservoir sampling. It maintains additions and deletions to and from the sample (*Anis & Nasir, 2018*).

**Sampling using Vapnik–Chervonenkis dimension**: TipTap algorithms *Nasir et al. (2021)*: try to maintain a uniform sample of connected k-vertex subgraphs with an neighborhood exploration procedure. The sample size is not fixed like rs, it is flexible and allows desired approximation quality. In defining sample complexity bounds, Vapnik–Chervonenkis dimension (*Riondato & Upfal, 2018*), a key concept from statistical learning theory is used and it helps to derive sufficient sample size as in the MaNIACS algorithm (*Preti, De Francisci Morales & Riondato, 2023*).

In this section, we started by giving some graph basics like the definition of a graph, subgraph, subgraph isomorphism, graph embedding, static graph, dynamic graph. In addition to basic graph definitions, increments as series of small graphs or as stream of nodes and edges on evolving graphs are explained each with an example. Then the subject processes of the survey like frequent subgraph mining, dynamic subgraph mining, graph sampling and approximate dynamic frequent subgraph mining are introduced. All of these are required background information to understand the algorithms surveyed and compared in the following sections of the article.

## Frequent subgraph mining algorithms for evolving graphs

In this section, we introduce the literature on frequent subgraph mining algorithms for dynamic graphs. We start with presenting a detailed comparison table where all related research is given with their general input/output attributes (input graph type, increment type, increment content and output type) and programming attributes (graph representation, data structure, algorithmic approach and base algorithm). We then discuss each algorithm introduced in the table. We wrap up the section with a discussion of subject algorithms.

Although the number of algorithms are not as many as the static counterparts, in dynamic frequent graph mining literature, various algorithms have also been proposed using different approaches and characteristics. Table 2 gives a comparison among these algorithms with their attributes. The first group of attributes in the table are related with the input and output of the algorithms whereas the second group of attributes are related with

the internal characteristics of the algorithms. The ***increments type,*** as the listed algorithms are incremental algorithms, the increments can be a series of small graphs or a stream of nodes and edges. The ***graph type*** of the input graphs can have one of the following possible types: Undirected labelled graphs (UL), undirected graphs (U), directed graphs (D), various kinds of graph data (V) or labeled graph (L). The ***increments*** can have Addition (A) or Deletion (D) of Nodes/Edges or Addition (A) of Edges. ***Output type*** indicates the essential purpose of each algorithm; it can be to find all frequent subgraphs or closed subgraphs.

The second group of attributes starts with the ***graph representation***, which is considered as one of the most effective attributes on the consumption of runtime and memory and can be adjacency matrix, adjacency list and canonical labelling. The ***data structure*** represents data structure that is used in the algorithm and it can be DFS tree, Suffix trees, DS-Tree, DS-Table, DS-Matrix or Index Structure. The ***algorithmic approach*** shows the frequent subgraph mining approach of the algorithm where the possible values can be Apriori or Pattern growth. ***Programming approach*** shows whether the algorithm utilize parallel programming or not. ***Base algorithm*** gives information related with static base algorithm since some incremental algorithms are developed based on extensions of the static ones.

The Dynamic GREW (*Borgwardt, Kriegel & Wackersreuther, 2006*) investigates how pattern mining on static graphs can be extended to time series of graphs. Specifically, it handles dynamic graphs with edge insertions and edge deletions over time. They define a frequency and provide algorithmic solutions for finding frequent dynamic subgraph patterns. Existing subgraph mining algorithms can be easily integrated into this framework to make them handle dynamic graphs. Experimental results in the paper on real-world data confirm the practical feasibility of proposed approach, and what the limitations of Dynamic GREW are; it assumes that the input dynamic graph has a fixed set of nodes, and edges are inserted and deleted over time. Also, there is an extra overhead to identify dynamic patterns. It misses some interesting patterns.

*Berlingerio & Bonchi (2009)* introduced Graph Evolution Rule Miner (GERM), a novel type of frequency based pattern that describe the evolution of large networks over time, at a local level. The input for this approach is a sequence of snapshots of an evolving graph; the main purpose is to mine the rules that describe the local changes in it. This approach uses the support based on minimum image to extract patterns which frequency is greater than a minimum support threshold. After that, graph-evolution rules are extracted from frequent patterns that satisfy a given minimum confidence constraint, the rules extraction framework is similar to it in classical rule mining. Experiments are done on four large real-world networks (two social networks, and two co-authorship networks), using different time granularities. Experiments approve feasibility and the utility of o framework. The limitations of GERM: it is designed for undirected graphs, nodes and edges are only added and never deleted. It assumed that node and edge labels do not change over time.

The algorithm in *Miyoshi, Ozaki & Ohkawa (2011)* handles the problem of mining frequent patterns and valid rules representing graph evolutions (structural changes) in a graph with time information. They propose an efficient technique for extracting representative patterns and rules; they use graph-based summarization of discovered rules. This done by using certain measures provided by the summary, so it is expected to find

Ergenç Bostanoğlu and Abuzayed (2024), *PeerJ Comput. Sci.*, DOI 10.7717/peerj-cs.2361

**Table 2 Algorithms for dynamic frequent subgraph mining.**

| Algorithm | Year | Increments type | Graph type | Increments | | | | Output type | Graph representation | Data structure | Algorithmic approach | Programming approach | Base algorithm |
|---|---|---|---|---|---|---|---|---|---|---|---|---|---|
| | | | | **A** | | **D** | | | | | | | |
| | | | | **Edge** | **Node** | **Edge** | **Node** | | | | | | |
| Dynamic GREW (*Borgwardt, Kriegel & Wackersreuther, 2006*) | 2006 | Small graphs | L | √ | | √ | | All frequent subgraphs | Suffix trees | Adjacency matrix | Apriori | Serial | Grew (*Kuramochi & Karypis, 2004*) |
| Germ (*Berlingerio & Bonchi, 2009*) | 2009 | Stream of nodes and edges | U | √ | √ | | | All frequent subgraphs | DFS tree | Canonical label | Pattern growth | Serial | Gspan *Yan & Han (2002)* |
| Time-evolving Graph (*Miyoshi, Ozaki & Ohkawa, 2011*) | 2011 | Stream of nodes and edges | U | √ | √ | | | All frequent subgraphs | Directed Acyclic Graph | Canonical label | Pattern growth | Serial | Germ |
| IncGraphMiner (*Bifet & Gavaldà, 2011*) | 2011 | Small graphs | V | √ | √ | | | Closed subgraphs | DFS tree | Canonical label | Pattern growth | Serial | CloseGraph (*Yan & Han, 2003*) |
| WinGraphMiner (*Bifet & Gavaldà, 2011*) | 2011 | Stream of nodes and edges | V | √ | √ | | | Closed subgraphs | DFS tree | Canonical label | Pattern growth | Serial | CloseGraph |
| AdaGraphMiner (*Bifet & Gavaldà, 2011*) | 2011 | Stream of nodes and edges | V | √ | √ | | | Closed subgraphs | DFS tree | Canonical label | Pattern growth | Serial | CloseGraph |
| span (*Lakshmi & Meyyappan, 2013*) | 2013 | Small graphs | UL | √ | √ | | | All frequent subgraphs | DFS tree | Adjacency list | Pattern growth | Parallel | Gspan |
| FP from dense graph streams (*Braun et al., 2014*) | 2014 | Small graphs | U | √ | | √ | | All frequent pattern | DS-Tree, DSTable, DSMatrix | Canonical label | Pattern growth | Serial | FP-Growth |
| IncGM+ (*Abdelhamid et al., 2017*) | 2017 | Stream of edges | D | √ | | √ | | All frequent subgraphs | Index structure | Canonical label | Pattern growth | Serial | StreamFsm, Moment (*Chi et al., 2004*) |
| edge-based FSM from graph streams (*Cuzzocrea et al., 2015*) | 2015 | Small graphs | U | √ | | √ | | All frequent subgraphs | DS-Tree, DSTable, DSMatrix | Canonical label | Pattern growth | Serial | FP from dense graph streams (*Braun et al., 2014*) |
| DyFSM (*Chen et al., 2023*) | 2023 | Small graphs | U | √ | | √ | | All frequent subgraphs | DFS Tree | Canonical label | Pattern growth | Serial | Gspan |

more interesting information which are difficult to be discovered by the traditional support and confidence measures. Proposed algorithm based on gSpan and Germ, it differs from gSpan that it handles single graph input and work on graph patterns have time points, it differs from Germ that proposed method that handles multi edges.

In *Bifet & Gavaldà (2011)*, the first work on close stream while only two frequent close graph algorithms on static graphs are introduced. Bifet et al. proposed new method for mining frequent closed subgraphs. The method is IncGraphMiner, it works on frequent weighted closed graph mining. this method works on coresets of closed subgraphs, compressed representations of graph sets, and maintain these sets in a batch-incremental manner; it handles the potential concept drift. The proposed algorithm is based on close graph, which takes time overhead to summarize the patterns.

In *Bifet & Gavaldà (2011)*, the first work on close stream, they proposed frameworks are for studying graph pattern mining on time-varying streams. Two new methods for mining frequent closed subgraphs are presented. The methods are WinGraphMiner and AdaGraphMiner, which work on frequent weighted closed graph mining. All methods work on corsets of closed subgraphs, compressed representations of graph sets, and maintain these sets in a batch-incremental manner but use different approaches to address potential concept drift. The above three algorithms are based on close graph which takes time overhead to summarize the patterns.

Most algorithms are programmed in serial approach while very few are developed in parallel manner like span (*Lakshmi & Meyyappan, 2013*). The span (*Lakshmi & Meyyappan, 2013*) is based on gSpan. It handles special class of undirected labelled simple graphs, graphs with unique or no labels. Parallel programming is used in order to decrease the complexity. Proposed solution, is applicable in cases where graph dataset fits in main memory. Two techniques, DFS lexicographic order and minimum DFS code are introduced in gSpan. With the help of these techniques, a novel canonical labeling system is constructed which supports DFS search. However, finding minimum DFS code is still an NP- complete problem. That is why a modified DFS representation is proposed. This representation utilizes advantages of gSpan, and allows parallel programming on multi-core processing technology to improve the performance. The number of duplicate graphs generated are slightly more than gSpan algorithm since sub graph mining from frequent single edge graphs are done in parallel.

The StreamFSM *Ray, Holder & Choudhury (2014)* discovers the frequent subgraphs in a graph, represented by a stream of labeled nodes and edges. In this model, updates to the graph arrive in the form of batches that contain new nodes and edges. The proposed method continuously reports the frequent subgraphs that are estimated to be found in the entire graph as each batch arrives. It is evaluated using five large dynamic graph datasets: the Hetrec 2011 challenge data, Twitter, DBLP and two synthetic datasets. It is evaluated against two popular static large graph miners, *i.e.,* SUBDUE and GERM. Experimental results show that it can find the same frequent subgraphs as a non-incremental approach applied to snapshot graphs, and in less time. The drawback of the StreamFSM algorithm: In terms of several parameters that have to be tuned in order to get the optimal performance in terms of time and accuracy/interestingness of results. Also, it assumes that we have

access to the entire graph as the graph grows. This assumption will not work in a real world-streaming scenario (*Ray, Holder & Choudhury, 2014*). In addition, it only handles increments with additions.

The IncGM+ *Abdelhamid et al. (2017)* is a fast incremental approach for continuous frequent subgraph mining on a single large evolving graph. It adapts the notion of "fringe" to the graph context, which is the set of subgraphs that are on the border between frequent and infrequent subgraphs. IncGM+ maintains fringe subgraphs and utilize them in the search space pruning. In order to increase the efficiency, an efficient index structure is proposed to maintain selected embeddings, with minimal memory overhead. These embeddings are employed to avoid subgraph isomorphism operations. Furthermore, the proposed system supports batch updates. Experiments are done using large real-world graphs, it verifies that IncGM+ outperforms existing algorithms by up to three orders of magnitude, scales to much larger graphs, and consumes less memory. The limitation of IncGM+ that it still needs to enumerate and track an exponential number of candidate subgraphs.

DyFSM *Chen et al. (2023)* is a frequent subgraph mining algorithm designed for dynamic databases where incremental updates and decremental updates are handled. It uses a buffer named Fringe- subgraph collection to keep these updates. A novel representation based on DFS code used by DyFSM allows enumeration of frequent subgraphs rapidly without subgraph isomorphism computations.

### Discussion on the algorithms

In this section dynamic frequent subgraph mining algorithms are presented. The section starts with a detailed comparison table (Table 2) where different characteristics of dynamic frequent subgraph mining algorithms are summarized. The first group of characteristics are related with the year and input and output type of the algorithm. The second group of characteristics details and compares internal characteristics of the algorithms. After this informative and comparative summary, we brief each algorithm.

Our findings related with exact dynamic frequent subgraph mining algorithms are:

- There is only one pioneer algorithm (Dynamic Grew) which is Apriori based. The rest of the exact dynamic frequent subgraph mining algorithms are Pattern growth based. The Apriori based algorithms (*Inokuchi, Washio & Motoda, 2000*) generate candidates using breadth first strategy (BFS) and then calculate frequencies of candidates by doing isomorphism checking. These candidate generation and calculation of frequencies phase requires multiple passes over the data and increases the complexity of the algorithm. On the other hand Pattern growth based algorithms (*Yan & Han, 2003*) generate candidates based on depth first strategy (DFS). Efficiency of the pattern growth approach comes with its elimination of candidate generation and subgraph isomorphism testing. The candidates are generated starting from minimal frequent subgraphs and then adding an edge at each iteration until they are still frequent.

- Common sought representation of subgraphs in these algorithms is canonical label (*Bifet & Gavaldà, 2011*; *Cuzzocrea et al., 2015*). This standard string representation allows handling the graph isomorphism issues.

- Most of the algorithms employ DFS tree (*Berlingerio & Bonchi, 2009*; *Chen et al., 2023*). In this domain DFS trees are efficient since the worst case complexity of traversing the whole tree is O(V+E) where V represents the number of vertices and E represents the number of edges in the graph.

- There is only one parallel algorithm span (*Lakshmi & Meyyappan, 2013*). This algorithm uses the advantage of multicore processor technology. The rest of the algorithms are single thread. This issue is mentioned in Research Opportunities section of this survey as well, as a potential research axis.

- One last thing about these algorithms; they are versions of some static subgraph mining algorithms like gspan (*Yan & Han, 2002*), Germ (*Berlingerio & Bonchi, 2009*) and CloseGraph (*Yan & Han, 2003*).

## Approximate frequent subgraph mining algorithms for evolving graphs

In this section, we will start by talking on the challenges of approximate dynamic subgraph mining. Then, like the previous section, we will introduce the literature on approximate frequent subgraph mining algorithms for dynamic graphs. We start with presenting a detailed comparison table where all related research is given with their general input/output attributes (input graph type, increment type and increment content), programming attributes (graph representation, data structure and objective) and sampling attributes (sampling strategy, statistical proof and data in the sample). We then brief each algorithm introduced in the table. We wrap up the section with a general summary of the facts of subject algorithms.

FSM task on large graphs is computationally challenging due to combinatorial explosion in possible number of vertex and vertex labels of subgraphs and subgraph isomorphism operations needed. This process is generally NP-complete and exact algorithms tend to scale poorly. Using sample of the graph whenever the approximate results serve the purpose is a solution. Another reason for using sampling is for the cases when the entire graph cannot be retrieved. Main challenge of sampling is to provide theoretical guarantees on the quality of the sample. In fact we see examples sampling based FSM algorithms in the context of static graphs like MaNIACS (*Preti, De Francisci Morales & Riondato, 2023*), Ap-FSM (*Bhatia & Rani, 2018*; *Purohit, Choudhury & Holder, 2017*; *Zou & Holder, 2010*) with or without sample quality guarantees.

We already introduced in background section, sampling methods used in the context of approximate dynamic frequent subgraph mining. In the remaining part of this section, we try to analyze sampling based FSM algorithms on evolving graphs. These approximate algorithms face challenge of maintaining good quality sample in addition to handling increments done to the graph and complexity of base FSM algorithms.

As already mentioned, approximate dynamic frequent subgraph mining algorithms work on the dynamic sample instead of dynamic graph and their results are approximate. They require a sampling method in order to keep dynamic and high quality sample of input graphs. Table 3 gives a comparison among these algorithms with their attributes. The first group of attributes in the table are related with general input and output of the algorithms whereas the second group of attributes are related with the internal characteristics of the

algorithms and third group of attributes denote the characteristics of the sampling scheme. The *increments type,* as the listed algorithms are incremental algorithms, the increments can be a series of small graphs or a stream of nodes and edges. The **graph type** of the input graphs can have one of the following possible types: various kinds of graph data (V), connected undirected graphs (CU) and (L) labeled. The *increments* can have nodes/edges where each column can have, Addition (A) or Deletion (D).

The second group of attributes starts with the **graph representation**, which is considered as one of the most effective attributes on the consumption of runtime and memory and can be DFS Tree or Array with hashed based index or Array with hash map. The **data structure** represents data structure that is used in the algorithm and it can be dictionary data structure or Canonical label. The *objective* of this group of attributes is to minimize execution time or maximize accuracy. The third group of attributes are related with sampling scheme of the algorithm where *sampling strategy* gives information if sample size is fixed or flexible. **Statistical proof** shows whether the algorithm uses an upper bound formula and unbiased estimator. In addition, *data in the sample* is either edges or subgraphs for the approximate algorithms listed.

The StreamFSM (*Ray, Holder & Choudhury, 2014*) discovers the frequent subgraphs in a graph, represented by a stream of labeled nodes and edges. In this model, updates to the graph arrive in the form of batches that contain new nodes and edges. The proposed method continuously reports the frequent subgraphs that are estimated to be found in the entire graph as each batch arrives. It is evaluated using five large dynamic graph datasets: the Hetrec 2011 challenge data, Twitter, DBLP and two synthetic datasets. It is evaluated against two popular static large graph miners and experimental results show that it can find the same frequent subgraphs as a non-incremental approach applied to snapshot graphs, and in less time. The drawback of the StreamFSM algorithm: (i) in terms of several parameters that have to be tuned in order to get the optimal performance in terms of time and accuracy/interestingness of results, (ii) it assumes that we have access to the entire graph as the graph grows however this assumption will not work in a real world-streaming scenario, (iii) it only handles increments with additions.

SR and OSR are two dynamic approximate algorithms (*Aslay et al., 2018*). Both algorithms use reservoir sampling technique (RS) as introduced in *Vitter (1985)*. RS is a fixed size randomized sampling technique; it maintains a fixed-size uniform sample of the data. The sample size is assigned as an input parameter. The algorithms starts and fills the reservoir until it reaches the user defined fixed size. Once the maximum sample size is reached or the reservoir is filled, each new item replaces an existing item in the sample. The addition of an edge affects only the subgraphs in the local neighborhoods up to specified neighborhood. A uniform sample of subgraphs is maintained by iterating through the subgraphs in the neighborhood of the newly inserted edge.

Triest algorithm (*De Stefani et al., 2016*) counts triangles in incremental streams with fixed memory size, it uses standard reservoir sampling (*Vitter, 1985*) to maintain the edge sample. The drawback of this algorithm that edge deletion from reservoir is done randomly when reservoir is full, by this random deletion some important patterns might be lost.

**Table 3  Approximate dynamic frequent subgraph mining algorithms.**

| Algorithm | General | | | | | | | Algorithm | | | Sampling | | |
|---|---|---|---|---|---|---|---|---|---|---|---|---|---|
| | Year | Increments type | Graph type | Increments | | | | Graph representation | Data structure | Objective | Sampling strategy | Statistical proof | Data in the sample |
| | | | | A | | D | | | | | | | |
| | | | | Edge | Node | Edge | Node | | | | | | |
| StreamFSM (*Ray, Holder & Choudhury, 2014*) | 2014 | Stream of nodes and edges | V | √ | √ | | | DFS Tree | Dictionary data structure | Minimize execution time, maximize accuracy/ interestingness | Neighborhood sampling | Uniform | Edges |
| Triest (*De Stefani et al., 2016*) | 2016 | Stream of edges | CU | √ | | √ | | Array with hash map | Canonical label | Minimize execution time | Reservoir sampling | | Edges |
| FSM in an evolving graph (*Aslay et al., 2018*) | 2018 | Stream of edges | CU | √ | | √ | | Array with hashed based index | Canonical label | Maximize accuracy | Reservoir sampling | | 3 subgraphs |
| TiTaps (*Nasir et al., 2021*) | 2021 | Stream of Edges | L | √ | | √ | | Array with hashed based index | Canonical label | Maximize accuracy | Sample with VC dimension | Unbiased frequency estimator | 3 or 4 subgraphs |

TipTap (*Nasir et al., 2021*) is suite of algorithms designed for mining frequent subgraphs in evolving graphs with high quality approximations. The objective is to maintain a uniform sample of k-vertex subgraphs with a neighborhood exploration procedure. Instead of fixed sample size of earlier algorithms the algorithms in the suit rely on flexible sample size where the bounds determined by Vapnik–Chervonenkis dimension (*Riondato & Upfal, 2018*). This allows deriving a sufficient sample size that is independent from the size of the graph.

### Discussion on the algorithms

In this section approximate dynamic frequent subgraph mining algorithms are explained. Similar to the previous section a detailed comparison table (Table 3) presenting different characteristics of approximate dynamic frequent subgraph mining algorithms is given. These characteristics are; year, increment type (small graphs/stream of edges and nodes), graph type (Labeled/Connected Undirected/Various), increments (Addition/Deletion of Nodes/Edges), graph representation (DFS Tree/Array with hash based index/Array with hash map) data structure (Dictionary data structure/Canonical label), objective (maximize accuracy/minimize execution time/maximize interestingness), sampling strategy (neighborhood sampling /reservoir sample/sample with VC dimension), statistical proof (uniform/unbiased frequency estimator) and data in the sample (edges/three or four subgraphs). After this informative and comparative summary, we briefly each mentioned algorithm.

Our findings related with approximate dynamic frequent subgraph mining algorithms are:

- Common representation of subgraphs in these types of algorithms is again canonical label (*Bifet & Gavaldà, 2011*; *Cuzzocrea et al., 2015*). This standard spring representation allows handling the graph isomorphism issues.
- Most of the algorithms in this case employ arrays with hash maps or hashed based index.
- Two of the algorithms (StreamFSM and Triest) keeps edges in the sample; the other two keep three or four subgraphs in the sample. The ones keeping edges have objective of minimizing execution time. The ones keeping three or four subgraphs in the sample have objective of maximizing accuracy.
- Two of the algorithms employ reservoir sampling strategy; where sample size is fixed; they try to avoid growing sample size. However, they cannot give statistical guarantees related with the accuracy of the results.
- A recent algorithm, TipTaps (*Nasir et al., 2021*) tries to maximize accuracy by keeping three or four subgraphs in the sample and adjust sample size according to desired accuracy at the same time.

## RESEARCH OPPORTUNITIES

Mining frequent subgraphs in dynamic graphs is very active field. There are many articles published each focusing on a special context and specific problem. Even so, there are many missing blocks and research opportunities. Let us try to point out these briefly:

- Efficient algorithms: Frequent subgraph discovery is a computationally expensive process even on static graphs. It becomes more challenging with evolving graphs since discovery of frequent subgraphs from scratch by the arrival of each increment is not practical. Instead of this frequent subgraph discovery algorithms should be devised as to handle the increments and return frequent subgraphs at any point of time. New and efficient algorithms are always welcome; especially the ones proposing novel data structures and clever pruning strategies. Data structures used by these algorithms should be able to maintain subgraphs with proper encodings that would also allow subgraph isomorphism check. Pruning strategies would serve the elimination of useless candidate subgraphs as early as possible and keep minimum potentially useful data.

- Sampling quality: A trend in recent years has been to consider approximate results instead of exact results whenever approximation serves the purpose for the sake of scalability. Therefore, we start seeing approximate frequent subgraph mining algorithms over evolving graphs, which use different sampling schemes as mentioned in this article. However, the main difficulty with those algorithms is the need of large sample size in order to increase the accuracy that is the rate of the frequent subgraphs discovered from the sample over frequent subgraphs really exist in the whole data. Defining sample bounds, guaranteeing good representative sample, limiting sample size, and keeping important subgraphs in the sample are the main challenges here. All these challenges are also valid and open issues for static approximate frequent subgraph mining algorithms. In the case of dynamic approximate frequent subgraph mining algorithms, these become even more challenging.

- Dynamic attributed graphs: Attributed graphs overcome the limitation of labeled graphs since one label for each vertex is not practical in many real life problems. So when working with attributed graphs, frequent subgraph mining problem needs to be extended as not only counting edges but also finding ways to incorporate varying attributes in mining process. Frequent subgraph mining over dynamic attributed graphs try to discover meaningful attribute evolution rules. Main challenge again is to reduce the search space since traditional anti-monotonic measures of frequent subgraph mining world is not applicable here. New measures for search space pruning and novel interestingness measures for credible attribute rule finding are required.

- Different models: Another interesting possibility is to consider dynamic graph models other than evolving graphs. In some real life applications like traffic control, dynamic graphs are in the form of sequence of snapshots. It means that time dimension is added to the model. Frequent subgraph mining process should consider that dimension.

- Constraints and interest: Another research direction of the field is extending pattern definitions or interesting measures. Many domains pose different constraints. Subgraph mining process is computationally expensive. Applying constraints after the mining process is additional cost. Therefore incorporating those constraints in mining process can increase efficiency of the whole process and result in more interesting subgraph discovery for the user. Measures of objective and subjective interestingness for different domains is a research opportunity as well.

- Parallel or distributed computation: Finding frequent subgraph patterns in a big graph is an important problem with many applications. Since this problem is NP-hard, parallel processing and novel approaches are needed to accelerate the mining process. However parallelization of the process brings out extra challenges as partitioning the data, workload management, minimization of inter process messages. Another issue here can be related with distributed graph data as an outcome of big data era. Mediation of the process over data sites and minimization of the communication cost might inspire further research.

- Domain specific applications: Novel applications of frequent subgraph mining over evolving graphs may raise new challenges and inspire further research.

## CONCLUSION

Frequent subgraph mining (FSM) is defined as discovering all the subgraphs in a graph that appear more than a given support threshold. Recurrent structures found by such a process can give important insights about the underneath graph data. Most of the FSM algorithms consist of two phases; candidate generation and frequency calculation. Avoiding generation of the same candidate more than once is an issue of the first candidate generation phase. In the frequency calculation phase, the challenge is isomorphism testing which is an NP-complete problem. FSM process with mentioned challenges become even more challenging with dynamic graphs. Such environments have additional challenges as to handle increments: increments in the form of addition/deletion of edges/nodes or as small graphs in our context. FSM algorithms need to be devised to discover frequent subgraphs at any point of time without re-mining the whole graph data. They need to have practical data structures to keep updated subgraphs of the whole graph data from which frequent subgraphs can be mined at any time. They need concise representation of the subgraphs as well.

When we look at the world of dynamic subgraph mining algorithms for evolving graphs in the scope of this survey, we see two main group of algorithms. The objective of the first group is to extract all frequent subgraphs or frequent closed subgraphs from the evolving graph at any point of time. On the other hand, the second group of these algorithms trade-off accuracy with execution time and try to find approximate frequent subgraphs. Second group facilitate a sampling scheme and by the help of the sample they try to find frequent subgraphs as close to their exact counterparts as possible. Their additional challenge is to maintain a sample, which is a smaller yet good representative of the whole graph.

Our findings related with the first group of algorithms-exact dynamic frequent subgraph mining algorithms; most of the algorithms (i) are pattern growth based to avoid the cost candidate generation and multiple scans of the data, (ii) use canonical label representation that allows handling the graph isomorphism issues, (iii) employ DFS tree since its traversal has linear complexity, (iv) do not take advantage of multicore processor technology and (v) are versions of some static subgraph mining algorithms.

Our findings related with the second group of algorithms-approximate dynamic frequent subgraph mining algorithms; (i) they are not as many as the first group, (ii) they use

canonical label again to handle graph isomorphism issues, (iii) they employ arrays with hash maps or hashed based index for efficient search, (iv) two of the algorithms (StreamFSM and Triest) keeps edges in the sample; the other two keep 3 or 4 subgraphs in the sample, (vi) the ones keeping edges have objective of minimizing execution time, the ones keeping 3 or 4 subgraphs in the sample have objective of maximizing accuracy, (vii) two of the algorithms employ reservoir sampling strategy that fixes the sample size, (viii) recent ones try to adjust sample size according to desired accuracy.

This survey starts with giving detailed background information related with graph basics, increment types of dynamic graphs, frequent subgraph mining, dynamic frequent subgraph mining and sampling strategies in the context of approximate dynamic frequent subgraph mining. Then it gives a qualitative literature survey on dynamic frequent subgraph mining algorithms designed for evolving graphs with detailed comparison of these algorithms with respect to their various features. In the third part of the survey, another qualitative literature survey can be found; this time on approximate dynamic subgraph mining algorithms again with respect to their various features. This time with additional features that include the ones related with their sampling strategy. Similar to the previous part, review contains a comparison of the algorithms in addition to the brief introduction of each. Finally research opportunities of the domain from our perspective are given. Main motivation is to propose a thorough and comparative panorama of frequent subgraph mining algorithms over evolving graphs for the researchers of the field.

### Funding
The authors received no funding for this work.

### Competing Interests
The authors declare there are no competing interests.

### Author Contributions
- Belgin Ergenç Bostanoğlu conceived and designed the experiments, performed the experiments, analyzed the data, prepared figures and/or tables, authored or reviewed drafts of the article, and approved the final draft.
- Nourhan Abuzayed conceived and designed the experiments, performed the experiments, analyzed the data, prepared figures and/or tables, authored or reviewed drafts of the article, and approved the final draft.

### Data Availability
   This is a literature review.

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
