# Peer review of "Dynamic frequent subgraph mining algorithms over evolving graphs: a survey"

_PeerJ Computer Science, doi:10.7717/peerj-cs.2361_

## Round 0.1 · original submission · Major Revisions

Dear authors,

Thank you for submitting your article. Feedback from the reviewers is now available. It is not recommended that your article be published in its current format. However, we strongly recommend that you address the issues raised by the reviewers and resubmit your paper after the necessary changes and additions. When submitting the paper it will be better to provide the outcomes of the review and meaningful takeaways for the reader. The future research directions should also be itemized and discussed in more detail. "In this chapter," should be corrected. English grammar and writing style errors should also be corrected.

Best wishes,

Reviewer 1 ·

Basic reporting

Summary
This paper provides a survey of FSM algorithms, with a particular focus on evolving graphs.

Strong Points
S1. The problem addressed is important in graph mining.
S2. Tables 2 and 3 offer a comprehensive overview of various FSM algorithms.

Weak Points
W1. The presentation and writing need significant improvement [D1 and D2].
W2. The paper lacks a detailed discussion of the proposed algorithms, such as what differentiates one algorithm from another.
W3. The paper does not thoroughly address the time complexity and efficiency of the proposed algorithms.

Detailed Comments
D1. some writing issues and typos:
- The abstract should be concise, brief, and direct.
- Line 39: The explanation of why graph mining is popular is inaccurate.
- Line 40: The use of 'etc.' is unnecessary.
- Line 168: Refer to 'section' rather than 'chapter.'
- Line 173: Correct the typo 'vertexes' to 'vertices.'

D2. The quality of the figures needs improvement, for example:
- Figure 1's caption is in blue, while other figures use different colors; the figure itself is pixelated.
- Figure 3 is missing.

Experimental design

no comment

Validity of the findings

no comment

Additional comments

NA

Cite this review as

Reviewer 2 ·

Basic reporting

Dynamic Frequent Subgraph Mining Algorithms over Evolving Graphs: A survey

Score: Major Revision.

Comments:
The paper provides a comprehensive survey on dynamic frequent subgraph mining algorithms, particularly focusing on evolving graphs. It outlines several key contributions. This paper introduces the differences between exact dynamic frequent subgraph mining algorithms and approximate frequent subgraph mining algorithms. Specifically, this paper outlines the sampling methods of the introduced algorithms. Overall, this paper gives a overview to the researchers who aim to work on the dynamic frequent subgraph mining algorithms on evolving graphs. I have the following comments for this paper:

1. This paper lists dynamic frequent subgraph mining algorithms on evolving graphs. However, there is no summary for the introduced works in this paper, either the overall comparison between the introduced works. Therefore, the authors are suggested to add a summary of the compared works.
2. More figures should be provided in this paper. Meanwhile, some figures are not properly displayed in the final PDF file, e.g., Figure 3. Since there are too few figures, it is hard to show the differences between the sampling methods, and the differences between dynamic frequent subgraph mining algorithms on evolving graphs and other graphs. Besides, the lack of figures also makes the paper hard to follow. It is strongly recommended to add more figures to show the process of different sampling methods and the technical differences between the algorithms.
3. More related works should be introduced. This paper focuses on dynamic frequent subgraph mining algorithms. However, some other background knowledge should also be introduced including subgraph matching [1,2,3,4], subgraph counting [5,6,7,8] and similarity all-matching [9].
4. In the Research Opportunities Section, I think the possibility of applying distributed computation on dynamic frequent subgraph mining can be discussed.


[1] Sun, Zhao, et al. "Efficient Subgraph Matching on Billion Node Graphs." Proceedings of the VLDB Endowment 5.9 (2012).
[2] Sun, Shixuan, and Qiong Luo. "In-memory subgraph matching: An in-depth study." Proceedings of the 2020 ACM SIGMOD International Conference on Management of Data. 2020.
[3] Qiao, Miao, Hao Zhang, and Hong Cheng. "Subgraph matching: on compression and computation." Proceedings of the VLDB Endowment 11.2 (2017): 176-188.
[4] Wang, Hanchen, et al. "Reinforcement learning based query vertex ordering model for subgraph matching." 2022 IEEE 38th International Conference on Data Engineering (ICDE). IEEE, 2022.
[5] Ribeiro, Pedro, et al. "A survey on subgraph counting: concepts, algorithms, and applications to network motifs and graphlets." ACM Computing Surveys (CSUR) 54.2 (2021): 1-36.
[6] Zhang, Hao, et al. "Distributed subgraph counting: a general approach." Proceedings of the VLDB Endowment 13.12 (2020): 2493-2507.
[7] Wang, Hanchen, et al. "Neural subgraph counting with Wasserstein estimator." Proceedings of the 2022 International Conference on Management of Data. 2022.
[8] Zhao, Kangfei, et al. "A learned sketch for subgraph counting." Proceedings of the 2021 International Conference on Management of Data. 2021.
[9] Zhu, Gaoping, et al. "TreeSpan: efficiently computing similarity all-matching". Proceedings of the 2012 International Conference on Management of Data. 2012.

Experimental design

NA

Validity of the findings

NA

Additional comments

NA

Cite this review as

---

## Round 0.2 · accepted · Accept

Dear authors,

One of the two reviewers refused to review the revised manuscript. The other referee accepted to review the revised manuscript but did not submit his/her comments within the normal time limit. When I examined your article, it is thought that the necessary additions and arrangements have been made according to the editor and reviewers' opinions and the article has been sufficiently improved. As such, the article is considered acceptable.

Best wishes,